# Improving Adversarial Defense with Self-supervised Test-time Fine-tuning

## Abstract

Although adversarial training and its variants currently constitute the most effective way to achieve robustness against adversarial attacks, their poor generalization limits their performance on the test samples. In this work, we propose to improve the generalization and robust accuracy of adversarially-trained networks via self-supervised test-time fine-tuning. To this end, we introduce a meta adversarial training method to find a good starting point for test-time fine-tuning. It incorporates the test-time fine-tuning procedure into the training phase and strengthens the correlation between the self-supervised and classification tasks. The extensive experiments on CIFAR10, STL10 and Tiny ImageNet using different self-supervised tasks show that our method consistently improves the robust accuracy under different attack strategies for both the white-box and black-box attacks.

## 1 Introduction

Adversarial training and its variants (Madry et al., 2018; Wang et al., 2019; Zhang et al., 2019) are currently recognized as the most effective defense mechanism against adversarial attacks. However, adversarial training generalizes poorly; the robust accuracy gap between the training and test set in adversarial training is much larger than the training-test gap in standard training of deep networks (Neyshabur et al., 2017; Zhang et al., 2017). Unfortunately, classical techniques to overcome overfitting in standard training, including regularization and data augmentation, only have little effect in adversarial training (Rice et al., 2020).

The poor generalization of adversarial training is a consequence of adversarial test examples lying far away from the training set (Huan et al., 2019). To adapt to these examples, we propose to fine-tune the network for each test mini-batch. Since the labels of the test images are not available, we exploit self-supervision, which is widely used in the standard training of networks (Chen et al., 2020b; Gidaris et al., 2018; He et al., 2020). Fine-tuning self-supervised tasks is a substitute of fine-tuning the classification loss at the inference time. Thus, minimizing this self-supervised loss function yields better generalization on the test set.

To make our test-time fine-tuning strategy effective, we need to search for a good starting point during training. As will be shown in our experiments, adversarial training does *not* provide the optimal starting point. We therefore formulate the search for the network parameters that can be effectively fine-tuned as a bilevel optimization problem. Specifically, we introduce a meta adversarial training strategy dedicated to our self-supervised fine-tuning inspired by the model-agnostic meta-learning (MAML) framework (Finn et al., 2017). To this end, we treat the classification of each batch of adversarial images as one task and minimize the corresponding classification error of the self-supervised fine-tuned network. This meta adversarial training strategy strengthens the correlation between the self-supervised and classification tasks so that self-supervised test-time fine-tuning further improves the robust accuracy.

In order to reliably evaluate our method, we follow the suggestions of (Tramer et al., 2020) and design an adaptive attack strategy that is fully aware of the test-time fine-tuning. We empirically demonstrate the effectiveness of our method on the commonly used CIFAR10 (Krizhevsky et al., 2009), STL10 (Coates et al., 2011) and Tiny ImageNet (Le & Yang, 2015) datasets under both standard (Andriushchenko et al., 2020; Croce & Hein, 2020a; Madry et al., 2018) and adaptive attacks in the scenarios of both white-box and black-box attacks. The experiments evidence that our method consistently improves the robust accuracy under all attacks.

Our contributions can be summarized as follows:

1. We introduce the framework of self-supervised test-time fine-tuning for adversarially-trained networks and show that it improves the robust accuracy of the test data.

2. We propose a meta adversarial training strategy based on the MAML framework to find a good starting point and strengthen the correlation between the self-supervised and classification tasks.

3. Following the suggestions of (Tramer et al., 2020), we design an adaptive attack strategy that is fully aware of our test-time fine-tuning. The experiments show that our approach is valid on diverse attack strategies, including standard and adaptive ones in both the white-box and black-box attacks.

Our code is available through this link.

## 2 RELATED WORK

**Adversarial Training.** In recent years, many approaches have been proposed to defend networks against adversarial attacks (Guo et al., 2018; Liao et al., 2018; Song et al., 2018). Among them, adversarial training (Madry et al., 2018) stands out as one of the most robust and popular methods, even under various strong attacks (Athalye et al., 2018; Croce & Hein, 2020a). Adversarial training optimizes the loss of adversarial examples to find parameters that are robust to adversarial attacks. Several variants of adversarial training have been proposed (Wang et al., 2019; Zhang et al., 2019), and their performance is similar to the original version of adversarial training (Rice et al., 2020).

One important problem that limits the robust accuracy of adversarial training is overfitting. Compared with training on clean images, the gap of robust accuracy between the training and test set is much larger in adversarial training (Rice et al., 2020). Moreover, traditional techniques to prevent overfitting, such as data augmentation, have little effect. Recently, some methods have attempted to flatten the weight loss landscape to improve the generalization of adversarial training. In particular, Adversarial Weight Perturbation (AWP) (Wu et al., 2020) achieves this by designing a double-perturbation mechanism that adversarially perturbs both inputs and weights. In addition, learning-based smoothing can flatten the landscape and improve the performance (Chen et al., 2021).

**Self-supervised Learning.** In the context of non-adversarial training, many self-supervised strategies have been proposed, such as rotation prediction (Gidaris et al., 2018), region/component filling (Criminisi et al., 2004), patch-base spatial composition prediction (Trinh et al., 2019) and contrastive learning (Chen et al., 2020b; He et al., 2020). While self-supervision has also been employed in adversarial training (Chen et al., 2020a; Kim et al., 2020; Yang & Vondrick, 2020; Hendrycks et al., 2019), their methods only use self-supervised learning at training time to regularize the parameters and improve the robust accuracy. By contrast, we propose to perform self-supervised fine-tuning at test time, which we demonstrate to significantly improve the robust accuracy on test images. As will be shown in the experiments, the self-supervised test-time fine-tuning has larger and complementary improvements over the training time self-supervision.

**Test-time Fine-tuning.** Test-time fine-tuning has been used in various fields, such as image super-resolution (Shocher et al., 2018) and domain adaption (Sun et al., 2020; Wang et al., 2021). While our work is thus closely related to Test-Time Training (TTT) in (Sun et al., 2020), we target a significantly different scenario. TTT assumes that all test samples have been subject to the same distribution shift compared to the training data. As a consequence, it *incrementally* updates the model parameters when receiving new test images. By contrast, in our scenario, there is no systematic distribution shift, and it is therefore more effective to fine-tune the parameters of the *original* model for every new test mini-batch. This motivates our meta adversarial training strategy, which searches for the initial model parameters that can be effectively fine-tuned in a self-supervised manner.

## 3 METHODOLOGY

We follow the traditional multi-task learning formulation (Caruana, 1997) and consider a neural network with a backbone $z = E(x; \theta_E)$ and $K + 1$ heads. One head $f(z; \theta_f)$ outputs the classification result while the other $K$ heads $g_1(z; \theta_{g1}), ..., g_K(z; \theta_{gK})$ correspond to $K$ auxiliary self-supervised

tasks. $\theta = (\theta_E, \theta_f, \theta_{g1}, \cdots, \theta_{gk})$ encompasses all trainable parameters, and we further define

$$F = f \circ E; \quad G_k = g_k \circ E, \ k = 1, 2, ..., K. \tag{1}$$

Furthermore, let $D = \{(x_1, y_1), (x_2, y_2), \cdots, (x_n, y_n)\}$ denote the training set containing $n$ images with corresponding labels, and $\widetilde{D} = \{(\widetilde{x}_1, \widetilde{y}_1), (\widetilde{x}_2, \widetilde{y}_2), \cdots, (\widetilde{x}_m, \widetilde{y}_m)\}$ be the test set containing $m$ images. For further illustration, the labels of the test set are shown. However, they are unknown to the networks at test time. We denote the adversarial examples of $x$ and $\widetilde{x}$ as $x^{adv}$ and $\widetilde{x}^{adv}$, respectively. They satisfy the constraints $\|x^{adv} - x\| \leq \varepsilon$ and $\|\widetilde{x}^{adv} - \widetilde{x}\| \leq \varepsilon$, and $\varepsilon$ is the size of the adversarial budget. For any set $S$, we represent the average loss over the elements in $S$ as

$$\mathcal{L}(S) = \frac{1}{|S|} \sum_{s_i \in S} \mathcal{L}(s_i) \tag{2}$$

where $|S|$ is the number of elements in $S$. We use $F(\cdot; \theta)$ and $\mathcal{L}(\cdot; \theta)$ to denote a network $F$ and a loss function $\mathcal{L}$ parametrized by $\theta$, but we often omit the symbol $\theta$ to simplify the notation. The general classification loss, such as the cross-entropy, is denoted by $\mathcal{L}_{cls}$. We use the superscript "AT" to denote the adversarial training loss. For example, we define

$$\mathcal{L}_{cls}^{AT}(S) = \frac{1}{|S|} \sum_{x_i, y_i \in S} \max_{\|x_i^{adv} - x_i\| \leq \varepsilon} \mathcal{L}_{cls}(F(x_i^{adv}), y_i) \ . \tag{3}$$

Because of the poor generalization of adversarial training, the adversarially-trained model may not give the correct label for a large portion of the adversarial test examples. That is, for a test sample $\widetilde{x}^{adv}$ with ground-truth label $\widetilde{y}$, we may have $\arg\max_j F(\widetilde{x}^{adv}; \theta)_j \neq \widetilde{y}$ , where $F(\widetilde{x}^{adv}; \theta)_j$ denotes the output probability of the $j$-th class. Since we do not know the label $\widetilde{y}$ at test time, we cannot directly optimize the classification loss $\mathcal{L}_{cls}(F(\widetilde{x}^{adv}), \widetilde{y})$. We therefore use self-supervision as a substitute to fine-tune the backbone $E$. We provide more details on our self-supervised test-time fine-tuning scheme in Section 3.1 and introduce our meta adversarial training method in Section 3.2. In Section 3.3, we discuss the specific self-supervised tasks used in the experiments, and further analyze our algorithm in Section 3.4.

## 3.1 TEST-TIME FINE-TUNING

Our goal is to perform self-supervised learning on the test examples to overcome the overfitting problem of adversarial training. To this end, let us suppose that an adversarially-trained network with parameters $\theta^0$ receives a mini-batch of $b$ adversarial test examples $\widetilde{B}^{adv} = \{(\widetilde{x}_1^{adv}, \widetilde{y}_1), (\widetilde{x}_2^{adv}, \widetilde{y}_2), \cdots, (\widetilde{x}_b^{adv}, \widetilde{y}_b)\}$ , As the labels $\{\widetilde{y}_i\}_{i=1}^b$ are not available, we propose to fine-tune the backbone parameters $\theta_E$ by optimizing the loss function

$$\mathcal{L}_{SS}(\widetilde{B}^{adv}) = \frac{1}{b} \sum_{k=1}^K \lambda_k \sum_{i=1}^b \mathcal{L}_{SS,k}(G_k(\widetilde{x}_i^{adv}); \theta_E, \theta_{gk}) \ , \tag{4}$$

which encompasses $K$ self-supervised tasks. Here, $\mathcal{L}_{SS,k}$ represents the loss function of the $k$-th task and $\{\lambda_k\}_{k=1}^K$ are the weights balancing the contribution of each task.

The number of images $b$ may vary from 1 to $m$. $b = 1$ corresponds to the online setting, where only one adversarial image is available at a time, and the backbone parameters $\theta_E$ are adapted to every new image. The online setting is the most practical one, as it does not make any assumptions on the number of adversarial test images the network receives. By contrast, $b = m$ corresponds to the offline setting where all adversarial test examples are available at once. It is similar to the transductive learning (Gammerman et al., 1998; Vapnik, 2013). Note that our online setting differs from the online test-time training described in TTT (Sun et al., 2020); we do not incrementally update the network parameters as new samples come, but instead initialize fine-tuning from the same starting point $\theta^0$ for each new test image.

Eqn (4) encourages $\theta_E$ to update in favor of the self-supervised tasks. However, as the classification head $f$ was only optimized for the old backbone $E(\cdot; \theta_E^0)$, it will typically be ill-adapted to the new parameters $\theta_E^*$, resulting in a degraded robust accuracy. Furthermore, for a small $b$, the model tends to overfit to the test data, reducing $\mathcal{L}_{SS}$ to 0 but extracting features that are only useful for the self-supervised tasks.

To overcome these problems, we add an additional loss function acting on the training data that both regularizes the backbone $E$ and optimizes the classification head $f$ so that $f$ remains adapted to the fine-tuned backbone $E(\cdot; \theta_E^*)$. Specifically, let $B \subset D$ denote a subset of the training set. We then add the regularizer

$$\mathcal{L}_R(B) = \mathcal{L}_{cls}^{AT}(B) = \frac{1}{|B|} \sum_{x_i, y_i \in B} \max_{\|x_i^{adv} - x_i\| \leq \varepsilon} \mathcal{L}_{cls}(F(x_i^{adv}), y_i) \quad (5)$$

to the fine-tuning process. In short, Eqn (5) evaluates the adversarial training loss on the training set to fine-tune the parameters $\theta_f$ of the classification head. It also forces the backbone $E$ to extract features that can be used to make correct prediction, i.e., to prevent $\theta_E$ from being misled by $\mathcal{L}_{SS}$ when $b$ is small.

Combining Eqn (4) and Eqn (5), our final test-time fine-tuning loss is

$$\mathcal{L}_{test}(\widetilde{B}^{adv}, B) = \mathcal{L}_{SS}(\widetilde{B}^{adv}) + \lambda \mathcal{L}_R(B) \quad (6)$$

where $\lambda$ sets the influence of $\mathcal{L}_R$. Algorithm 1 describes our test-time self-supervised learning algorithm in detail. As SGD is more efficient for more large amount of data, we use SGD to optimize $\theta$ when $b$ is large (*e.g.* offline setting). This version of algorithm is deferred to the Appendix B.

---

**Algorithm 1** Self-supervised Test-time Fine-tuning

---

**Input:** Initial parameters $\theta^0$; Adversarial test images $\widetilde{B}^{adv} = \{\widetilde{x}_i^{adv}\}_{i=1}^b$; Training data $D$; Learning rate $\eta$; Steps $T$; Weights $\lambda_k$ and $\lambda$
**Output:** Prediction of $\widetilde{x}_i^{adv}$: $\widehat{y}_i$
1: **for** $t = 1$ to $T$ **do**
2:     Sample a batch of training images $B \subset D$
3:     Find adversarial $x_i^{adv}$ of training image $x_i \in B$ by PGD attack.
4:     Calculate $\mathcal{L}_{test}$ in Eqn (6)
5:     $\theta^t = \theta^{t-1} - \eta \nabla_{\theta^{t-1}} \mathcal{L}_{test}(\widetilde{B}^{adv}, B; \theta^{t-1})$
6: **end for**
7: **return** Prediction $\widehat{y}_i = \arg\max_j F(\widetilde{x}_i^{adv}; \theta^T)_j$

---

### 3.2 META ADVERSARIAL TRAINING

To make the best out of optimizing $\mathcal{L}_{test}$ at test time, we should find suitable starting point $\theta^0$, i.e., a starting point such that test-time self-supervised learning yields better robust accuracy. We translate this into a meta learning scheme, which entails a bilevel optimization problem.

Specifically, we divide the training data into $s$ small exclusive subsets $D = \cup_{j=1}^s B_j$ and let $B_j^{adv}$ to be adversaries of $B_j$. We then formulate meta adversarial learning as the bilevel minimization of

$$\mathcal{L}_{meta}(D; \theta) = \frac{1}{s} \sum_{B_j \subset D} \mathcal{L}_{cls}^{AT}(B_j; \theta_j^*(\theta)), \quad \theta_j^* = \arg\min_\theta \mathcal{L}_{SS}(B_j^{adv}; \theta), \quad (7)$$

where $\mathcal{L}_{SS}$ is the self-supervised loss function defined in Eqn (4) and $\mathcal{L}_{cls}^{AT}$ is the loss function of adversarial training defined in Eqn (3). As bilevel optimization is time-consuming, following MAML (Finn et al., 2017), we use a single gradient step of the current model parameters $\theta$ to approximate $\theta_j^*$. That is, we compute

$$\theta_j^* \approx \theta - \alpha \nabla_\theta \mathcal{L}_{SS}(B_j^{adv}; \theta) . \quad (8)$$

In essence, this meta adversarial training scheme searches for a starting point such that fine-tuning with $\mathcal{L}_{SS}$ will lead to a good robust accuracy. If this holds for all training subsets, then we can expect the robust accuracy after fine-tuning at test time also to increase. Note that, because the meta learning objective of Eqn (7) already accounts for classification accuracy, the regularization by $\mathcal{L}_R$ is not needed during meta adversarial learning.

**Accelerating Training.** To compute the gradient $\nabla_\theta \mathcal{L}_{meta}(D; \theta)$, we need to calculate the time-consuming second order derivatives $-\alpha \nabla_\theta^2 \mathcal{L}_{SS}(B_j^{adv}; \theta) \nabla_{\theta_j^*} \mathcal{L}_{cls}^{AT}(B_j; \theta_j^*)$ . Considering that adversarial training is already much slower than standard training (Shafahi et al., 2019), we cannot

afford another significant training overhead. Fortunately, as shown in (Finn et al., 2017), second order derivatives have little influence on the performance of MAML. We therefore ignore them and take the gradient to be

$$\nabla_\theta \mathcal{L}_{meta}(D; \theta) \approx \frac{1}{s} \sum_{B_j \subset D} \nabla_{\theta_j^*} \mathcal{L}_{cls}^{AT}(B_j; \theta_j^*) . \tag{9}$$

However, by ignoring the second order gradient, only the parameters on the forward path of the classifier $F$, *i.e.*, $\theta_E$ and $\theta_f$, will be updated. In other words, optimizing Eqn (7) in this fashion will not update $\{\theta_{gk}\}_{k=1}^K$. To nonetheless encourage each self-supervised head $G_k$ to output the correct prediction, we incorporate an additional loss function encoding the self-supervised tasks,

$$\mathcal{L}_{SS}^{AT}(D) = \sum_k \lambda_k \mathcal{L}_{SS,k}^{AT}(D) = \sum_k \frac{\lambda_k}{|D|} \sum_{x_i \in D} \max_{\|x_i^{adv} - x_i\| \leq \varepsilon} \mathcal{L}_{SS,k}(G_k(x_i^{adv})) . \tag{10}$$

Note that we use the adversarial version of $\mathcal{L}_{SS}$ to provide robustness to the self-supervised tasks, which, as shown in (Chen et al., 2020a; Hendrycks et al., 2019; Yang & Vondrick, 2020), is beneficial for the classifier. The final meta adversarial learning objective therefore is

$$\mathcal{L}_{train}(D) = \mathcal{L}_{meta}(D) + \lambda' \mathcal{L}_{SS}^{AT}(D) \tag{11}$$

where $\lambda'$ balances the two losses. Algorithm 2 shows complete meta adversarial training algorithm.

---

**Algorithm 2** Meta Adversarial Training

---

**Input:** Training set $D$; Learning rate $\alpha, \beta$; Iterations $T$; Weights $\lambda_k$ and $\lambda'$
**Output:** Starting parameters $\theta_0$ for the test-time fine-tuning
1: **for** $t = 1$ to $T$ **do**
2:     Sample $q$ exclusive batches of training images $B_1, B_2, \cdots, B_q \subset D$
3:     Using PGD to find the adversaries $B_j^{adv}$: $x_{j,i}^{adv} = \arg\max_{\|x_{j,i}^{adv} - x_{j,i}\| \leq \varepsilon} \mathcal{L}_{cls}(F(x_{j,i}^{adv}), y_{j,i})$
4:     **for** batches $B_1, B_2, \cdots, B_q$ **do**
5:         $\theta_j^* = \theta - \alpha \nabla_\theta \mathcal{L}_{SS}(B_j^{adv}; \theta)$
6:         $l_{meta,j} = \mathcal{L}_{cls}^{AT}(B_j; \theta_j^*)$
7:     **end for**
8:     $\theta = \theta - \frac{\beta}{q} \sum_{B_j} \left[ \nabla_{\theta_j^*} l_{meta,j} + \lambda' \nabla_\theta \mathcal{L}_{SS}^{AT}(B_j; \theta) \right]$
9: **end for**
10: **return** Trained parameters $\theta^0 = \theta$

---

### 3.3 SELF-SUPERVISED TASKS

In principle, any self-supervised tasks can be used for test-time fine-tuning, as long as they are positively correlated with the robust accuracy. However, for the test-time fine-tuning to remain efficient, we should not use too many self-supervised tasks. Furthermore, as we aim to support the fully online setting, where only one image is available at a time, we cannot incorporate a contrastive loss (Chen et al., 2020b; He et al., 2020; Kim et al., 2020) to $\mathcal{L}_{SS}$. In our experiments, we therefore use two self-supervised tasks that have been shown to be useful to improve the classification accuracy: **Rotation Prediction** and **Vertical Flip Prediction**.

**Rotation Prediction** is a widely used self-supervision task proposed in (Gidaris et al., 2018) and has been employed in adversarial training as an auxiliary task to improve the robust accuracy (Chen et al., 2020a; Hendrycks et al., 2019). Following (Gidaris et al., 2018), we create 4 copies of the input image by rotating it with $\Omega = \{0°, 90°, 180°, 270°\}$. The task then consists of a 4-way classification problem, where the head $g_{\text{rotate}}$ aims to predict the correct rotation angle. The loss for an image $x$ is the average cross-entropy over the 4 copies, given by

$$\mathcal{L}_{\text{rotate}}(x) = -\frac{1}{4} \sum_{\omega \in \Omega} \log(G_{\text{rotate}}(x_\omega)_\omega) , \tag{12}$$

where $x_\omega$ is the rotated image with angle $\omega \in \Omega$, $G_{\text{rotate}} = g_{\text{rotate}} \circ E$ denotes the classifier for rotation prediction, and $G_{\text{rotate}}(\cdot)_\omega$ is the predicted probability for the $\omega$ angle. The head $g_{\text{rotate}}$ is a fully-connected layer followed by a softmax layer.

Table 1: The statistics $\rho(\widetilde{x}^{adv})$ and two self-supervised tasks. The dataset is CIFAR10 and the network is WideResNet-34-10. Adversarial budget $\varepsilon = 0.031$

| Tasks | $\mathbb{E}(\rho(\widetilde{x}^{adv}))$ | $P(\rho(\widetilde{x}^{adv}) > 0)$ |
|---|---|---|
| Rotation | 0.15 | 68.51% |
| VFlip | 0.22 | 72.16% |

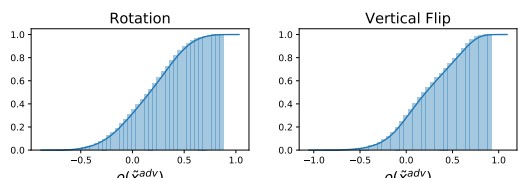

Figure 1: Empirical *cdf* of $\rho(\widetilde{x}_i^{adv})$ on CIFAR10 and WideResNet-34-10. Adversarial budget $\varepsilon = 0.031$

**Vertical Flip (VFlip) Prediction** is a self-supervised task similar to rotation prediction and has also been used for self-supervised learning (Saito et al., 2020). In essence, we make two copies of the input image and flip one copy vertically. The head $g_{\text{vflip}}$ then contains a 2-way fully-connected layer followed by a softmax layer and predicts whether the image is vertically flipped or not. The corresponding loss for an image $x$ is

$$\mathcal{L}_{\text{vflip}}(x) = -\frac{1}{2} \sum_{v \in V} \log(G_{\text{vflip}}(x_v)_v) \, , \tag{13}$$

where $V = \{\text{flipped}, \text{not flipped}\}$ is the operation set and $G_{\text{vflip}} = g_{\text{vflip}} \circ E$. $x_v$ denotes and transformed input and $G_{\text{vflip}}(\cdot)_v$ is the probability of operation $v$. Note that we do not flip the image horizontally as it is a common data augmentation technique and classifiers typically seek to be invariant to horizontal flip.

### 3.4 METHOD ANALYSIS

We now analyze why our test-time self-supervised fine-tuning improves the robust accuracy. To this end, we observe the correlation between the gradient of self-supervised loss $\mathcal{L}_{SS}$ and the classification loss $\mathcal{L}_{cls}$. For an input $\widetilde{x}_i^{adv}$ with label $\widetilde{y}_i$, the gradient correlation with respect to the shared parameters $\theta_E$ can be expressed as

$$\rho(\widetilde{x}_i^{adv}) = \frac{\nabla_{\theta_E}\mathcal{L}_{cls}(\widetilde{x}_i^{adv}, \widetilde{y}_i)^T \nabla_{\theta_E}\mathcal{L}_{SS}(\widetilde{x}_i^{adv})}{\|\nabla_{\theta_E}\mathcal{L}_{cls}(\widetilde{x}_i^{adv}, \widetilde{y}_i)\|_2 \|\nabla_{\theta_E}\mathcal{L}_{SS}(\widetilde{x}_i^{adv})\|_2} \, . \tag{14}$$

If $\rho(\widetilde{x}_i^{adv})$ is significantly larger than 0, then gradient descent w.r.t. $\mathcal{L}_{SS}$ should act as a good substitute for optimizing $\mathcal{L}_{cls}$. Let us consider a small step $\eta$ in the direction of $\nabla_{\theta_E}\mathcal{L}_{SS}(\widetilde{x}_i^{adv})$. Then, the classification loss $\mathcal{L}_{cls}$ can be approximated by Taylor expansion as

$$\begin{aligned}
&\mathcal{L}_{cls}(\widetilde{x}_i^{adv}, \widetilde{y}_i; \theta_E - \eta\nabla_{\theta_E}\mathcal{L}_{SS}(\widetilde{x}_i^{adv})) - \mathcal{L}_{cls}(\widetilde{x}_i^{adv}, \widetilde{y}_i; \theta_E) \\
&\approx -\eta\rho(\widetilde{x}_i^{adv})\|\nabla_{\theta_E}\mathcal{L}_{cls}(\widetilde{x}_i^{adv}, \widetilde{y}_i)\|_2 \|\nabla_{\theta_E}\mathcal{L}_{SS}(\widetilde{x}_i^{adv})\|_2 \, .
\end{aligned} \tag{15}$$

As $\theta_E$ contains millions of parameters, the gradient norm is typically large, even a small $\eta$ will decrease the classification loss when $\rho(\widetilde{x}_i^{adv})$ is significantly larger than 0.

Below, we further confirm this empirically. We choose cross-entropy as the classification loss. For all adversarial test inputs $\widetilde{x}^{adv} \sim \widetilde{D}^{adv}$, where $\widetilde{D}^{adv}$ is the adversaries of the test data, we regard $\rho(\widetilde{x}^{adv})$ as a random variable and calculate its empirical statistics on the test dataset. Table 1 shows the empirical statistics of an adversarially-trained model on CIFAR10, and Figure 1 shows the cumulative distribution of $\rho(\widetilde{x}^{adv})$. The mean of $\rho(\widetilde{x}^{adv})$ is indeed significantly larger than 0 and $P(\rho(\widetilde{x}^{adv}) > 0)$ is larger than the robust accuracy of the adversarially-trained network (50%-60%), which implies that self-supervised test-time fine-tuning helps to correctly classify the adversarial test images.

## 4 ADAPTIVE ATTACKS

In the white-box attacks, the attacker knows every detail of the defense method. Therefore, we need to assume that the attacker is aware of our test-time fine-tuning method and will adjust its strategy for generating adversarial examples accordingly. Here, we discuss one such strong adaptation strategy targeted to our method.

Suppose that the attacker is fully aware of the hyperparameters for test-time fine-tuning. Then, finding adversaries $\widetilde{B}^{adv}$ of the clean subset $\widetilde{B}$ can be achieved by maximizing the adaptive loss

$$\widetilde{x}_i^{adv} = \underset{\|\widetilde{x}_i^{adv} - x_i\| \leq \varepsilon}{\arg\max} \mathcal{L}_{attack}(F(\widetilde{x}_i^{adv}), y; \theta^T(\widetilde{B}^{adv})) \,, \tag{16}$$

where $\mathcal{L}_{attack}$ refers to the general attack loss, such as the cross-entropy or the difference of logit ratio (DLR) (Croce & Hein, 2020a). Let $\theta^T$ be the fine-tuned test-time parameters using Algorithm 1. At the $k$-th step of the attack, it depends on the input $\widetilde{B}^{(k)} = \{(\widetilde{x}_j^{(k)}, \widetilde{y}_j)\}_{j=1}^{b}$ via the update

$$\theta^{t+1} = \theta^t - \eta \nabla_{\theta^t} \mathcal{L}_{test}(\widetilde{B}^{(k)}, B) \,, \tag{17}$$

where $\mathcal{L}_{test}$ and $B$ are the loss function and subset of training images mentioned in Eqn (6). As $\theta^T$ is a function of the input $\widetilde{B}^{(k)}$, we can calculate the end-to-end gradient of $\widetilde{x}_i^{(k)} \in \widetilde{B}^{(k)}$ as $\nabla_{\widetilde{x}_i^{(k)}} \mathcal{L}_{attack}(F(\widetilde{x}_i^{(k)}); \theta^T(\widetilde{B}^{(k)}))$. However, $\theta^T$ goes through $T$ gradient descent steps, and thus calculating the gradient $\nabla_{\widetilde{x}_i^{(k)}} \theta^T(\widetilde{B}^{(k)})$ requires $T$-th order derivatives of the backbone $E$, which is virtually impossible if $T$ or the dimension of $\theta_E$ is large. We therefore approximate the gradient as

$$\text{Grad}(\widetilde{x}_i^{(k)}) \approx \nabla_{\widetilde{x}_i^{(k)}} \mathcal{L}_{attack}(F(\widetilde{x}_i^{(k)}); \theta^T) \,, \tag{18}$$

which treats $\theta^T$ as a fixed variable so that high-order derivatives from $\theta^T(\widetilde{B}^{(k)}(\widetilde{x}_i^{(k)}))$ can be avoided. Although this approximation makes $\text{Grad}(\widetilde{x}_i^{(k)})$ inaccurate, common white-box attacks use projected gradients, which are robust to such inaccuracies. For example, PGD only uses the sign of the gradient under an $\ell_\infty$ adversarial budget. Note that solving the maximization in Eqn (16) does not necessarily require calculating the gradient $\text{Grad}(\widetilde{x}_i^{(k)})$. For instance, we will also use Square Attack (Andriushchenko et al., 2020), a strong score-based black-box attack, to maximize Eqn (16) and generate adversaries for $\widetilde{B}$.

As another approximation to save time, one can also fixing $\theta^T$ for several iterations. This leverages the intuition that attack strategies often make small changes to the input $\widetilde{x}$, and thus, for the intermediate images in the $k$-th and ($k$+1)-th steps, $\theta^T(\widetilde{B}^{(k)})$ and $\theta^T(\widetilde{B}^{(k+1)})$ should be close. Therefore, a general version of our adaptive attacks only updates $\theta^T$ every $u$ iterations, with $u$ a hyperparameter. We provide the algorithms for PGD, AutoPGD, FAB and Square Attack in the Appendix C.

## 5 EXPERIMENTS

**Experimental Settings.** Following previous works (Cui et al., 2020; Huang et al., 2020), we consider $\ell_\infty$-norm attacks with an adversarial budget $\varepsilon = 0.031(\approx 8/255)$. We evaluate our method on three datasets: CIFAR10 (Krizhevsky et al., 2009), STL10 (Coates et al., 2011) and Tiny ImageNet (Le & Yang, 2015). We also use two different network architectures: WideResNet-34-10 (Zagoruyko & Komodakis, 2016) for CIFAR10, and ResNet18 (He et al., 2016) for STL10 and Tiny ImageNet. Following the common settings in adversarial training, we train the network for 100 epochs using SGD with a momentum factor of 0.9 and a weight decay factor of $5 \times 10^{-4}$. The learning rate $\beta$ starts at 0.1 and is divided by a factor of 10 at the 50-th and the 75-th epochs. The step size $\alpha$ in Eqn (8) the same as $\beta$; the factor $\lambda'$ in Eqn (11) is 1.0. We set the weight of each self-supervised task $\lambda_k = \frac{1}{K}$ where $K$ is the number of tasks. We use 10-iteration PGD with step size $0.007(\approx 0.031/4)$ to find the adversarial $B_j^{adv}$ during training. To prevent the computational burden of meta adversarial training, we use divide the classification of 32 images as one task and sample 8 mini-batches $B_1, ..., B_8$ in each iteration.(*i.e.* $|B_j| = 32$ and $q = 8$ in Algorithm 2). More detailed experimental settings are provided in the Appendix B.

**Attack Methods.** We consider four common white-box and black-box attack methods: PGD-20 (Madry et al., 2018), AutoPGD (Croce & Hein, 2020a), FAB (Croce & Hein, 2020b) and Square Attack (Andriushchenko et al., 2020). We apply both the standard and adaptive versions of these methods. Furthermore, for AutoPGD, we use both the cross-entropy and DLR (Croce & Hein, 2020a) loss. More details are provided in the Appendix C.

**Baselines.** We compare our method with the following methods: 1) Regular adversarial training (Regular AT), which uses $\mathcal{L}_{cls}^{AT}$ in Eqn (3). 2) Regular adversarial training with an additional self-supervised loss, i.e., using $\mathcal{L}_{cls}^{AT} + \lambda' \mathcal{L}_{SS}^{AT}$ for adversarial training, where $\mathcal{L}_{SS}^{AT}$ is given in Eqn (10).

Table 2: Robust test accuracy on CIFAR10 of the test-time fine-tuning on both the online setting and the offline setting. We use the WideResNet-34-10 with an $\ell_\infty$ budget $\varepsilon = 0.031$. AT means adversarial training and FT stands for fine-tuning. We underline the accuracy of the strongest attack and highlight the highest accuracy among them.

| Tasks | Methods | Square Attack | | PGD-20 | | AutoPGD | | FAB | |
|---|---|---|---|---|---|---|---|---|---|
| | | Standard | Adaptive | Standard | Adaptive | Standard | Adaptive | Standard | Adaptive |
| None | Regular AT | 62.51% | - | 55.74% | - | 52.14% | - | _51.34%_ | - |
| Rotation | Regular AT w/o FT | 63.54% | - | 56.64% | - | 52.57% | - | _51.87%_ | - |
| | Meta AT w/o FT | 63.96% | - | 57.35% | - | _53.09%_ | - | _53.09%_ | - |
| | Online FT | 65.52% | 65.85% | 59.52% | 59.50% | 57.93% | _56.96%_ | 75.58% | 77.69% |
| | Offline FT | 67.05% | 65.75% | 61.17% | 59.71% | 58.77% | **_57.63%_** | 78.12% | 68.60% |
| VFlip | Regular AT w/o FT | 62.09% | - | 55.50% | - | 52.79% | - | _51.24%_ | - |
| | Meta AT w/o FT | 66.15% | - | 59.73% | - | 53.41% | - | _53.02%_ | - |
| | Online FT | 66.91% | 66.16% | 61.47% | 59.40% | 58.74% | _56.79%_ | 75.68% | 80.57% |
| | Offline FT | 67.23% | 65.60% | 61.82% | 59.69% | 59.26% | **_58.06%_** | 75.60% | 72.24% |
| Rotation + VFlip | Regular AT w/o FT | 65.64% | - | 59.19% | - | 53.16% | - | _53.05%_ | - |
| | Meta AT w/o FT | 65.75% | - | 59.51% | - | 53.99% | - | _53.85%_ | - |
| | Online FT | 67.34% | 66.80% | 61.79% | 60.46% | 59.23% | _57.70%_ | 76.39% | 79.80% |
| | Offline FT | 68.50% | 66.05% | 62.87% | 60.54% | 60.25% | **_58.26%_** | 76.89% | 71.58% |

Table 3: STL10 results of test-time fine-tuning. We use a ResNet18 with an $\ell_\infty$ budget $\varepsilon = 0.031$. We underline the accuracy of the strongest attack and highlight the highest accuracy among them.

| Tasks | Methods | Square Attack | | PGD-20 | | AutoPGD | | FAB | |
|---|---|---|---|---|---|---|---|---|---|
| | | Standard | Adaptive | Standard | Adaptive | Standard | Adaptive | Standard | Adaptive |
| None | Regular AT | 44.83% | - | 37.89% | - | _35.78%_ | - | 35.64% | - |
| Rotation + VFlip | Regular AT w/o FT | 44.00% | - | 36.92% | - | _33.72%_ | - | 33.73% | - |
| | Meta AT w/o FT | 44.75% | - | 38.66% | - | 35.60% | - | _35.38%_ | - |
| | Online FT | 45.07% | 46.19% | 40.31% | 40.24% | _39.53%_ | 40.85% | 51.25% | 51.08% |
| | Offline FT | 47.86% | 48.03% | 45.21% | 43.33% | 43.78% | **_43.20%_** | 58.49% | 54.13% |

This corresponds to the formulation of (Hendrycks et al., 2019). 3) Meta adversarial learning (Algorithm 2) without test-time fine-tuning.

We include additional figures, experiments with different adversarial budget and experiments about label leaking in Appendix A.

## 5.1 ONLINE TEST-TIME FINE-TUNING

Let us first evaluate our method in the online setting, where only one adversarial image is available when fine-tuning the network. We fine-tune the network for 10 steps with a momentum factor of 0.9 and a learning rate $\eta = 5 \times 10^{-4}$. We set $\lambda_k = \frac{1}{K}$ and $\lambda = 15.0$.

**CIFAR10.** Table 2 shows the robust accuracy for different attacks and using two different tasks for fine-tuning. The adaptive attack is not applicable to models without fine-tuning. As we inject different self-supervised tasks into the adversarial training stage, and as different self-supervised tasks may impact the robust accuracy differently (Chen et al., 2020a), the robust accuracy without fine-tuning still varies. The vertical flipping task yields better robust accuracy before fine-tuning but its improvements after fine-tuning is small. By contrast, rotation prediction achieves low robust accuracy before fine-tuning, but its improvement after fine-tuning is the largest. Using both tasks together combines their effect and yields the highest overall accuracy after test-time fine-tuning.

Note that our self-supervised test-time fine-tuning, together with meta adversarial learning, consistently improves the robust accuracy under different attack methods. Under the strongest adaptive AutoPGD, test-time fine-tuning using both tasks achieves a robust accuracy of 57.70%, significantly outperforming regular adversarial training.

**STL10 and Tiny ImageNet.** As using both the rotation and vertical flip prediction led to the highest overall accuracy on CIFAR10, we focus on this strategy for STL10 and Tiny ImageNet. Table 3 and 4 shows the robust accuracy on STL10 and Tiny ImageNet using a ResNet18. Our approach also significantly outperforms regular adversarial training on these datasets.

Table 4: Tiny ImageNet results of test-time fine-tuning. We use a ResNet18 with an $\ell_\infty$ budget $\varepsilon = 0.031$. We underline the accuracy of the strongest attack and highlight the highest accuracy among them.

| Tasks | Methods | Square Attack | | PGD-20 | | AutoPGD | | FAB | |
|---|---|---|---|---|---|---|---|---|---|
| | | Standard | Adaptive | Standard | Adaptive | Standard | Adaptive | Standard | Adaptive |
| None | Regular AT | 28.5% | - | 20.6% | - | 17.5% | - | 17.2% | - |
| Rotation + VFlip | Regular AT w/o FT | 29.5% | - | 22.2% | - | 17.1% | - | 16.7% | - |
| | Meta AT w/o FT | 29.3% | - | 23.1% | - | 16.9% | - | 16.8% | - |
| | Online FT | 30.2% | 30.2% | 24.0% | 23.2% | 18.9% | 18.1% | 33.7% | 31.6% |
| | Offline FT | 32.4% | 31.0% | 25.6% | 24.1% | 23.7% | **20.6%** | 36.5% | 27.7% |

Table 5: Ablation study on the online test-time fine-tuning. The dataset is CIFAR10 and the task is the "Rotation + VFlip". All attacks are standard attacks. Removing the $\mathcal{L}_{SS}$ or $\mathcal{L}_R$ results in lower robust accuracy than the full method. SA stands for Square Attack.

| Methods | SA | PGD-20 | AutoPGD | FAB |
|---|---|---|---|---|
| Before FT | 65.75% | 59.51% | 53.99% | 53.85% |
| Online FT | **67.34%** | **61.79%** | **59.23%** | **76.39%** |
| Removing $\mathcal{L}_R$ | 66.83% | 60.45% | 57.32% | 75.63% |
| Removing $\mathcal{L}_{SS}$ | 65.44% | 60.24% | 55.64% | 75.08% |

Table 6: Ablation study on the online test-time fine-tuning. The dataset is CIFAR10 and the task is the "Rotation + VFlip". All attacks are standard attacks. SA stands for Square Attack.

| Methods | SA | PGD-20 | AutoPGD | FAB |
|---|---|---|---|---|
| Regular AT | 65.64% | 59.19% | 53.16% | 53.05% |
| Online FT | 66.26% | 60.18% | 56.86% | 75.26% |
| Improvement | 0.62% | 0.99% | 3.70% | 22.21% |
| Meta AT | 65.75% | 59.51% | 53.99% | 53.85% |
| Online FT | 67.34% | 61.79% | 59.23% | 76.39% |
| Improvement | **1.59%** | **2.28%** | **5.24%** | **22.54%** |

## 5.2 OFFLINE TEST-TIME FINE-TUNING

Let us now study the offline setting, where all test adversarial examples are available at test time. We use SGD to fine-tune the model for 10 epochs. We set $\lambda = 10.0$ and keep the other settings the same as for online fine-tuning. As shown in Table 2, 3, 4, the offline fine-tuning further improves the robust accuracy over the online version.

## 5.3 ABLATION STUDY

**Removing $\mathcal{L}_{SS}$ or $\mathcal{L}_R$.** In our previous experiments, test-time fine-tuning was achieved using a combination of two loss functions: $\mathcal{L}_{SS}$ and $\mathcal{L}_R$. To study the effect of each of these terms separately, we remove either one of them from $\mathcal{L}_{test}$. In Table 5, we report the robust accuracy after online fine-tuning using only $\mathcal{L}_R$ and only $\mathcal{L}_{SS}$. While, as expected, removing $\mathcal{L}_{SS}$ tends to reduce more accuracy than removing $\mathcal{L}_R$. It shows the benefits of our self-supervised test-time fine-tuning strategy. Nevertheless, the best results are obtained by exploiting both loss terms.

**Removing Meta Adversarial Training.** Our meta training strategy in Algorithm 2 aims to strengthen the correlation between the self-supervised tasks and classification. To show its effectiveness, we perform an ablation study where we fine-tune the model with regular adversarial training (*i.e.*, setting $\alpha = 0$ in the line 5 of Algorithm 2). We then perform the same test-time fine-tuning on the model without meta adversarial training, using the same hyperparameters as in the meta adversarial training case. As shown in Table 6, the robust accuracy and the improvements of fine-tuning are consistently worse without meta adversarial training.

## 6 CONCLUSION

We propose self-supervised test-time fine-tuning on adversarially-trained models to improve their generalization ability. Furthermore, we introduce a meta adversarial training strategy to find a good starting point for our self-supervised fine-tuning process. Our extensive experiments on CIFAR10, STL10 and Tiny ImageNet demonstrate that our method consistently improves the robust accuracy under different attack strategies, including strong adaptive attacks where the attacker is aware of our test-time fine-tuning technique. In these experiments, we utilize three different sources of self-supervision: rotation prediction, vertical flip prediction and the ensemble of them. While all settings lead to improved robust accuracy, some self-supervised tasks seem to have more impact than others. Finding better self-supervised tasks and their combinations will be the focus of our future work.

## REPRODUCIBILITY STATEMENT

We provide pesudocode of our algorithms and the detailed hyperparameters in the main text and Appendix. The datasets we use (CIFAR10, STL10 and Tiny ImageNet) are all publicly available. In addition, we provide the code to reproduce the experiment results here.

## ETHICS STATEMENT

Adversarial examples bring security threats to the current machine learning models. Adversarial examples may be maliciously used to attack the safety-critical systems like autonomous driving and face verification. While adversarial training is effective in defending against adversarial attacks, it suffers from poor generalization. We propose to use the self-supervised test-time fine-tuning to increase the robust accuracy of these models. It helps build safer machine learning systems.

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

# A  ADDITIONAL EXPERIMENTS

**Robust Accuracy v.s Fine-tuning Steps.** Figure 2 shows the robust accuracy at each step of the test-time fine-tuning for different self-supervised tasks and attack methods. When using the standard version of attacks, the robust accuracy gradually increases as fine-tuning proceeds. When using our adaptive attacks, the adversarial examples are generated to attack the network with $\theta^T$ ($T = 10$) instead of $\theta^0$. Thus, when the parameters gradually change from $\theta^0$ to $\theta^T$, the accuracy drops.

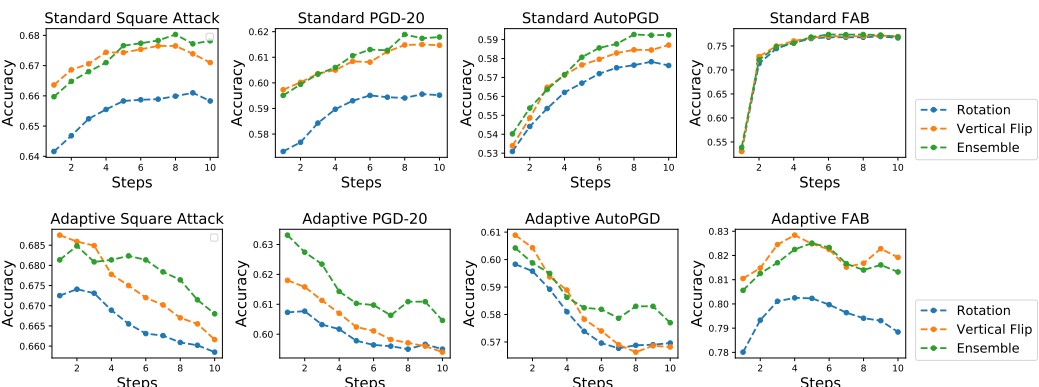

Figure 2: Robust accuracy at different steps of the online test-time fine-tuning on CIFAR10.

**Accuracy Improvement on Clean Images.** Our model is able to improve not only the robust accuracy but also the natural accuracy of clean images on adversarially-trained models. To evidence this, we maintain all the components of our model and simply replace the adversarial input images with clean images (*i.e.* replacing $\widetilde{B}^{adv}$ with clean inputs $\widetilde{B}$ in Algorithm 1) and perform the same self-supervised test-time fine-tuning.

As shown in Table 7, our approach increases the clean image accuracy. This phenomenon further strengthens our conjecture that the improvement of robust accuracy is due to the improvement of generalization instead of perturbing the model parameters, because the randomly perturbing the parameters usually lower the natural accuracy of the model. This also suggests that our method is applicable to the broader problem of improving the accuracy of deep learning models with poor generalization ability.

Table 7: Accuracy on clean images. Networks are trained with corresponding meta adversarial training.

| Methods | Rotation | VFlip | Rotation + VFlip |
|---|---|---|---|
| Without FT | 84.77% | 86.55% | 86.36% |
| Online FT | **86.10%** | **87.00%** | **87.10%** |

**Accuracy Improvement on Inputs with Different Adversarial Budget.** Our method is also able to improve the robust accuracy of inputs with different adversarial budgets. As shown in Table 8, we set $\ell_\infty$ budget of the adversarial inputs to be $0.015$ to perform the online test-time fine-tuning. The robust accuracy is ed improved.

Table 8: Robust test accuracy on CIFAR10 of the online test-time fine-tuning. We use the same WideResNet-34-10 as in Table 2, which is trained with $\ell_\infty$ budget $0.031$. The inputs are in the $\ell_\infty$ ball of $\varepsilon = 0.015$. The self-supervised task is the ensemble of rotation and vertical flip.

| Methods | Square Attack | | PGD-20 | | AutoPGD | | FAB | |
|---|---|---|---|---|---|---|---|---|
| | Standard | Adaptive | Standard | Adaptive | Standard | Adaptive | Standard | Adaptive |
| Meta AT w/o FT | 78.01% | - | 75.34% | - | 72.72% | - | 72.58% | - |
| Online FT | 80.50% | 79.87% | 77.14% | 76.75% | 77.25% | **74.93%** | 82.04% | 83.76% |

**Label Leaking.** One potential concern about our method is whether the effect of label leaking leads to the improvement of the robust accuracy. The attacker uses the ground truth label to generate the adversarial examples. Someone may worry it leaks the information of the ground truth label to the adversarial images and the test-time fine-tuning can utilize the information to guess the correct label.

Previous experiments on clean images already show that test-time fine-tuning is effective even if there is no information of the ground truth label. We further resolve the concern of the label leaking by the following experiments. The attacker randomly lowers the score of the false label to perform the adversarial attack. And we conduct the self-supervised test-time fine-tuning on these "adversarial" images. If our method uses the information of leaked label to improve the robust accuracy, it will predict the false label and reduce the accuracy. However, as shown in Table 9, the robust accuracy is also improved after the test-time fine-tuning. It demonstrates that the improvement of robust accuracy is not a result of label leaking. It is worth noting that such attack is not targeted at the ground truth label. Therefore, the accuracy of the input images, along with the accuracy improvement of the test-time fine-tuning, are similar to the clean images in Table 7.

Table 9: Experiments to rule out the possibility of label leaking. We use the WideResNet-34-10 trained with $\ell_\infty$ budget $\varepsilon = 0.031$ and show the robust test accuracy on CIFAR10 of the online test-time fine-tuning. The self-supervised task is the ensemble of rotation and vertical flip.

| Methods | Square Attack | | PGD-20 | | AutoPGD | | FAB | |
|---|---|---|---|---|---|---|---|---|
| | Standard | Adaptive | Standard | Adaptive | Standard | Adaptive | Standard | Adaptive |
| Meta AT w/o FT | 85.43% | - | 85.60% | - | 85.00% | - | 86.22% | - |
| Online FT | 86.56% | 87.63% | 86.29% | 86.68% | 86.10% | 85.90% | 87.61% | 86.97% |

**Comparison with SOAP (Shi et al., 2021).** Our method is different from SOAP as we are fine-tuning the model to adapt to new examples instead of purifying the input. We apply SOAP-RP to the adversarially-trained model and find that its improvement is marginal. Under AutoPGD, the accuracy is improved from 53.09% to 53.57%. This improvement is much smaller than our method, whose improvement is from 53.09% to 57.93%. SOAP only has little effect when combining with the commonly-used adversarial training.

**Transfer Attack.** In Table 10, we perform a transfer attack from the static adversarial defense. We use the robust networks with the same architecture as the substitute model, and the test-time fine-tuning also improves the robust accuracy.

Table 10: Accuracy on transfer attack on CIFAR10.

| Methods | Rotation | VFlip | Rotation + VFlip |
|---|---|---|---|
| Without FT | 84.77% | 86.55% | 86.36% |
| Online FT | **86.10%** | **87.00%** | **87.10%** |

**Expectation Attack.** In Table 11, we show the results of the expectation attack. We modify the adaptive attack and average the gradient from 10 fine-tuned models, whose training batches are different. We evaluate the model using the ensemble of rotation and vertical flip as the self-supervised task on CIFAR10. We evaluate the model with Adaptive-AutoPGD-EOT and Adaptive-SquareAttack-EOT. One is the strongest attack in our method and the other is a black-box attack that is less likely to be affected by gradient masking. The experiment shows that the expectation attack has little influence on the improvement of our test-time fine-tuning.

Table 11: Accuracy on expectation attack on CIFAR10 with the ensemble of rotation and vertical flip task.

| Attacks | w/o Fine-tuning | w/ Fine-tuning |
|---|---|---|
| Adaptive-AutoPGD | 53.99% | 57.70% |
| Adaptive-AutoPGD-EOT | 53.99% | 57.65% |
| Adaptive-SquareAttack | 65.75% | 66.80% |
| Adaptive-SquareAttack-EOT | 65.75% | 66.73% |

**Boundary Attack.** We use one of the SOTA decision-based attacks: RayS (Chen & Gu, 2020). We test it on CIFAR10 with the ensemble of rotation and vertical flip. Table 12 shows that our method also improves the robust accuracy for the decision-based attack.

Table 12: Accuracy on RayS on CIFAR10 with the ensemble of rotation and vertical flip task.

| Attacks | w/o Fine-tuning | w/ Fine-tuning |
|---|---|---|
| RayS | 65.61% | 77.38% |
| Adaptive-RayS | - | 75.03% |

**Combination with (Gowal et al., 2020).** We combine our test-time fine-tuning with the adversarial training using additional data (Gowal et al., 2020). We apply our Meta AT to it with the ensemble of rotation and vertical flip. Using a WideResNet-28-10, it achieves robust accuracy 62.07% under AutoPGD. With our test-time fine-tuning, the robust accuracy is improved to 64.34%. The improvement of robust accuracy is 2.27%.

**Inference Time.** Table 13 shows the inference time for different methods. While the inference time for our method is larger than SOAP and normal method when the batch size is 1, the inference time gets closer when using larger batch size. And the batch size of 20 or more is a common scenario of the inference.

Table 13: Average inference time for each instance using different methods.

| Batch Size | 1 | 5 | 10 | 20 | 40 |
|---|---|---|---|---|---|
| Normal | 17.1ms | 14.5ms | 13.2ms | 12.8ms | 11.7ms |
| SOAP (Shi et al., 2021) | 163ms | 91.2ms | 75.3ms | 73.1ms | 72.5ms |
| Ours | 545ms | 168ms | 118ms | 83.9ms | 82.9ms |

## B  DETAILS OF OUR EXPERIMENTAL SETTING

**Meta Adversarial Training.** We consider an $\ell_\infty$ norm with an adversarial budget $\varepsilon = 0.031$. We also use two different network architectures: WideResNet-34-10 for CIFAR10 and ResNet18 for STL10 and Tiny ImageNet. Following the common settings for adversarial training, we train the network for 100 epochs using SGD with a momentum factor of 0.9 and a weight decay factor of $5 \times 10^{-4}$. The learning rate $\beta$ starts at 0.1 and is divided by a factor of 10 after the 50-th and again after the 75-th epochs. The step size $\alpha$ in Eqn (8) is the same as $\beta$. The factor $\lambda'$ in Eqn (11) is set to 1.0. We use 10-iteration PGD (PGD-10) with a step size of 0.007 to find the adversarial image $B_j^{adv}$ at training time. The weight of each self-supervised task is set to $\lambda_k = \frac{1}{K}$. We set $|B_j| = 32$ and sample 8 batches $B_1, ..., B_8$ in each iteration. Furthermore, we save the model after the 51-st epoch for further evaluation, as the model obtained right after the first learning rate decay usually yields the best performance (Rice et al., 2020).

We use PGD with the standard cross-entropy loss to generate adversarial examples at training time in the line 3, line 6 and line 8 of Algorithm 2. The hyperparameters of the attacks are as follows:

- Line 3: PGD-10 with step size 0.007.
- Line 6: As $\theta_j^*$ is similar to $\theta$, the adversarial examples at this step are similar to those at Line 4. To save training time, we therefore choose the starting point of the attack as the adversarial examples in Line 4 and use PGD-2 with a step size of 0.005.
- Line 8: PGD-3 with step size 0.02.

**Online Test-time Fine-tuning.** We fine-tune the network for $T = 10$ steps with a momentum of 0.9 and a learning rate of $\eta = 5 \times 10^{-4}$. We set $\lambda_k = \frac{1}{K}$ and $\lambda = 15.0$. In line 2 of Algorithm 1, we sample a batch $B \subset D$ containing 20 training images. In line 3, we use PGD-10 with a step size of 0.007.

**Offline Test-time Fine-tuning.** The algorithm for offline fine-tuning is shown in Algorithm 3. As we stochastic gradient descent is more efficient for large amount of data, we use stochastic gradient

descent in the offline fine-tuning. This is the main different between Algorithm 1 (online fine-tuning) and Algorithm 3 (offline fine-tuning). We also fine-tune the network for 10 epochs. The batch size of each $\widetilde{B}_j^{adv}$ is 128. The other hyperparameters are the same as in the online version.

---

**Algorithm 3** Self-supervised Test-time Fine-tuning with SGD

---

**Input:**

    Initial parameters $\theta^0$; Adversarial test images $\widetilde{B}^{adv} = \{\widetilde{x}_i^{adv}\}_{i=1}^b$; Training data $D$; Learning rate $\eta$; Steps $T$; Weights $\lambda_k$ and $\lambda$

**Output:** Prediction of $\widetilde{x}_i^{adv}$: $\hat{y}_i$

 1: **for** $t = 1$ to $T$ **do**
 2:    Divide $\widetilde{B}^{adv}$ into $r$ subsets $\widetilde{B}_1^{adv}, ..., \widetilde{B}_r^{adv}$
 3:    **for** $\widetilde{B}_j^{adv}$ in $\widetilde{B}_1^{adv}, ..., \widetilde{B}_r^{adv}$ **do**
 4:        Sample a batch of training images $B \subset D$
 5:        Find adversarial $x_i^{adv}$ of training image $x_i \in B$ by PGD attack.
 6:        $\theta^t = \theta^{t-1} - \eta \nabla_{\theta^{t-1}} \mathcal{L}_{test}(\widetilde{B}_j^{adv}, B; \theta^{t-1})$
 7:    **end for**
 8: **end for**
 9: **return** Prediction $\hat{y}_i = \arg\max_j F(\widetilde{x}_i^{adv}; \theta^T)_j$

---

**Attacks.** The detailed settings of each attack are provided below:

- PGD-20. We use 20 iterations of PGD with step size $\gamma = 0.003$. The attack loss is the cross-entropy.

- AutoPGD. We use both the cross-entropy and the difference of logits ratio (DLR) as the attack loss. The hyperparameters are the same as in (Croce & Hein, 2020a).

- FAB. We use the code from (Croce & Hein, 2020a) and keep the hyperparameters the same.

- Square Attack. We set $T = 2000$ and the initial fraction of the elements $p = 0.3$. The other hyperparameters are the same as in (Andriushchenko et al., 2020).

For the adaptive versions, we set the interval $u = \lceil T/5 \rceil$.

## C   ALGORITHM FOR ADAPTIVE ATTACKS

In Algorithm 4, 6, 5 and 7, we show the algorithms for $\ell_\infty$ norm-based adaptive PGD, AutoPGD, Square Attack and FAB, respectively. The main difference between the original and adaptive version is the target loss function for maximization. The reader may refer to (Andriushchenko et al., 2020; Croce & Hein, 2020a;b) for more detailed description of the steps in these algorithms (*e.g.*, the condition for decreasing the learning rate in AutoPGD).

---

**Algorithm 4** $\ell_\infty$ Norm Adaptive PGD Attack

---

**Input:** Test images $\widetilde{B} = \{(\widetilde{x}_i, \widetilde{y}_i)\}$; Attack loss $\mathcal{L}_{attack}$; Step size $\gamma$; Iterations $T$; Intervals $u$; Adversarial budget $\varepsilon$; Trained parameters of the network $\theta^0$.
**Output:** Adversarial images $\widetilde{B}^{adv} = \{\widetilde{x}_i^{adv}\}$
1: Add random noise to $\widetilde{x}_i$ in $\widetilde{B}$ and get $\widetilde{B}'$
2: **for** $t = 1$ to $T$ **do**
3:    **if** $t \bmod u = 0$ **then**
4:       Get final parameters $\theta^T$ by taking $\widetilde{B}'$ as input image for Algorithm 1: $\theta = \theta^T$
5:    **end if**
6:    **for** $\widetilde{x}_i'$ in $\widetilde{B}'$ **do**
7:       $\text{Grad}(\widetilde{x}_i') = \nabla_{\widetilde{x}_i'} \mathcal{L}_{attack}(F(\widetilde{x}_i'), \widetilde{y}_i; \theta)$
8:       $\widetilde{x}_i' = \text{Clip}_{[\widetilde{x}_i - \varepsilon, \widetilde{x}_i + \varepsilon]}(\widetilde{x}_i' + \gamma \text{Sign}(\text{Grad}(\widetilde{x}_i')))$
9:    **end for**
10: **end for**
11: **return** Adversarial image $\widetilde{x}_i^{adv} = \widetilde{x}_i'$

---

---

**Algorithm 5** $\ell_\infty$ Norm Adaptive AutoPGD

---

**Input:** Test images $\widetilde{B} = \{(\widetilde{x}_i, \widetilde{y}_i)\}$; Attack loss $\mathcal{L}_{attack}$; Step size $\gamma$; Iterations $T$; Intervals $u$; Adversarial budget $\varepsilon$; Parameter of the adversarially-trained network $\theta^0$; Decay iterations $W = \{w_0, ..., w_n\}$; Momentum $\xi$
**Output:** Adversarial image $\widetilde{B}^{adv} = \{\widetilde{x}_i^{adv}\}$
1: Get final parameter $\theta^T$ by taking $\widetilde{B}$ as input image for Algorithm 1.
2: $\theta = \theta^T$
3: **for** $\widetilde{x}_i$ in $\widetilde{B}$ **do**
4:    $\widetilde{x}_i^0 = \widetilde{x}_i$
5:    $\text{Grad}(\widetilde{x}_i) = \nabla_{\widetilde{x}_i} \mathcal{L}_{attack}(F(\widetilde{x}_i), \widetilde{y}_i; \theta)$
6:    $\widetilde{x}_i^1 = \text{Clip}_{[\widetilde{x}_i - \varepsilon, \widetilde{x}_i + \varepsilon]}(\widetilde{x}_i^0 + \gamma \text{Sign}(\text{Grad}(\widetilde{x}_i)))$
7:    $l_i^0 = \mathcal{L}_{attack}(F(\widetilde{x}_i^0), \widetilde{y}_i; \theta)$
8:    $l_i^1 = \mathcal{L}_{attack}(F(\widetilde{x}_i^1), \widetilde{y}_i; \theta)$
9:    $l_i^* = \max\{l_i^0, l_i^1\}$
10:   $\widetilde{x}_i^* = \widetilde{x}_i^0$ **if** $l_i^* = l_i^0$ **else** $\widetilde{x}_i^* = \widetilde{x}_i^1$
11: **end for**
12: **for** $t = 1$ to $T - 1$ **do**
13:    **if** $t \bmod u = 0$ **then**
14:       Get final parameter $\theta^T$ by taking $\widetilde{B}^* = \{\widetilde{x}_i^*\}$ as input image for Algorithm 1.
15:       $\theta = \theta^T$
16:    **end if**
17:    **for** $i = 1, ..., |\widetilde{B}|$ **do**
18:       $\text{Grad}(\widetilde{x}_i^t) = \nabla_{\widetilde{x}_i^t} \mathcal{L}_{attack}(F(\widetilde{x}_i^t), \widetilde{y}_i; \theta)$
19:       $z_i^{t+1} = \text{Clip}_{[\widetilde{x}_i - \varepsilon, \widetilde{x}_i + \varepsilon]}(\widetilde{x}_i^t + \gamma \text{Sign}(\text{Grad}(\widetilde{x}_i^t)))$
20:       $\widetilde{x}_i^{t+1} = \text{Clip}_{[\widetilde{x}_i - \varepsilon, \widetilde{x}_i + \varepsilon]}(\widetilde{x}_i^t + \xi(z_i^{t+1} - z_i^t) + (1 - \xi)(\widetilde{x}_i^t - \widetilde{x}_i^{t-1}))$
21:       $l_i^{t+1} = \mathcal{L}_{attack}(F(\widetilde{x}_i^{t+1}), \widetilde{y}_i; \theta)$
22:       $\widetilde{x}_i^* = \widetilde{x}_i^{t+1}$ and $l_i^* = l_i^{t+1}$ **if** $l_i^{t+1} > l_i^*$
23:       **if** $k \in W$ and satisfy the condition of dropping learning rate **then**
24:          $\gamma = \gamma/2$ and $\widetilde{x}_i^{t+1} = \widetilde{x}_i^*$
25:       **end if**
26:    **end for**
27: **end for**
28: **return** Adversarial image $\widetilde{x}_i^{adv} = \widetilde{x}_i^*$

---

---

**Algorithm 6** $\ell_\infty$ Norm Adaptive Square Attack

---

**Input:** Test images $\widetilde{B} = \{(\widetilde{x}_i, \widetilde{y}_i)\}$; Attack loss $\mathcal{L}_{attack}$; Step size $\gamma$; Iterations $T$; Intervals $u$; Image size $w$; Color channels $c$; Adversarial budget $\varepsilon$; Parameter of the adversarially-trained network $\theta^0$.
**Output:** Adversarial image $\widetilde{B}^{adv} = \{\widetilde{x}_i^{adv}\}$
 1: Add noise to $\widetilde{x}_i$ in $\widetilde{B}$ and get $\widetilde{B}'$
 2: **for** $t = 1$ to $T$ **do**
 3:    **if** $t \bmod u = 0$ **then**
 4:       Get final parameter $\theta^T$ by taking $\widetilde{B}'$ as input image for Algorithm 1.
 5:       $\theta = \theta^T$
 6:    **end if**
 7:    **for** $\widetilde{x}_i'$ in $\widetilde{B}'$ **do**
 8:       $h^t \leftarrow$ side length of the square to modify (according to some schedule)
 9:       $\delta \leftarrow$ array of zeros of size $w \times w \times c$
10:       Sample uniformly $r, s \in \{0, ..., w - h^t\} \subset \mathbb{N}$
11:       **for** $j = 1, ..., c$ **do**
12:          $\rho \leftarrow \text{Uniform}(-2\varepsilon, 2\varepsilon)$
13:          $\delta_{r+1:r+h^t, s+1:s+h^t} = \rho \cdot 1_{h^t \times h^t}$
14:       **end for**
15:       $\widetilde{x}_i^{\text{new}} = \text{Clip}_{[\widetilde{x}_i - \varepsilon, \widetilde{x}_i + \varepsilon]}(\widetilde{x}_i' + \delta)$
16:       $l_i^{\text{new}} = \mathcal{L}_{attack}(F(\widetilde{x}_i^{\text{new}}), \widetilde{y}_i; \theta)$
17:       **if** $l_{\text{new}} < l^*$ **then**
18:          $\widetilde{x}_i' = \widetilde{x}_i^{\text{new}}$
19:          $l_i^* = l_i^{\text{new}}$
20:       **end if**
21:    **end for**
22: **end for**
23: **return** Adversarial image $\widetilde{x}_i^{adv} = \widetilde{x}_i'$

---

---

**Algorithm 7** $\ell_\infty$ Norm Adaptive FAB

---

**Input:** Test images $\widetilde{B} = \{(\widetilde{x}_i, \widetilde{y}_i)\}$; Step size $\gamma$; Iterations $T$; Intervals $u$; Adversarial budget $\varepsilon$; Trained parameters of the network $\theta^0$; $\alpha_{\max}$, $\eta$, $\beta$.

**Output:** Adversarial images $\widetilde{B}^{adv} = \{\widetilde{x}_i^{adv}\}$

1: Add random noise to $\widetilde{x}_i$ in $\widetilde{B}$ and get $\widetilde{B}'$
2: $v = +\infty$
3: **for** $t = 1$ to $T$ **do**
4:    **if** $t \bmod u = 0$ **then**
5:       Get final parameters $\theta^T$ by taking $\widetilde{B}'$ as input image for Algorithm 1: $\theta = \theta^T$
6:    **end if**
7:    **for** $\widetilde{x}_i'$ in $\widetilde{B}'$ **do**
8:       $\mathrm{Grad}(\widetilde{x}_i')_l = \nabla_{\widetilde{x}_i'} F(\widetilde{x}_i'; \theta)_l$
9:       $s = \arg\min_{l \neq y_i} \frac{|F(\widetilde{x}_i'; \theta)_l - F(\widetilde{x}_i'; \theta)_{y_i}|}{\|\mathrm{Grad}(\widetilde{x}_i')_l - \mathrm{Grad}(\widetilde{x}_i')_{y_i}\|_1}$
10:       $\delta^t = \mathrm{proj}_\infty(\widetilde{x}_i', \pi_s, C)$
11:       $\delta_{\mathrm{orig}}^t = \mathrm{proj}_\infty(\widetilde{x}_i, \pi_s, C)$
12:       $\alpha = \min\left\{\frac{\|\delta^t\|_\infty}{\|\delta^t\|_\infty + \|\delta_{\mathrm{orig}}^t\|_\infty}, \alpha_{\max}\right\} \in [0, 1]$
13:       $\widetilde{x}_i' = \mathrm{proj}_C\left((1-\alpha)(\widetilde{x}_i' + \eta\delta^t) + \alpha(\widetilde{x}_i + \eta\delta_{\mathrm{orig}}^t)\right)$
14:       **if** $\widetilde{x}_i'$ is not classified as $y_i$ **then**
15:          **if** $\|\widetilde{x}_i' - \widetilde{x}_i\|_\infty < v$ **then**
16:             $\widetilde{x}_i^{adv} = \widetilde{x}_i'$
17:             $v = \|\widetilde{x}_i' - \widetilde{x}_i\|_\infty$
18:          **end if**
19:          $\widetilde{x}_i' = (1-\beta)\widetilde{x}_i + \beta\widetilde{x}_i'$
20:       **end if**
21:    **end for**
22: **end for**
23: **return** Adversarial image $\widetilde{x}_i^{adv}$

---

