# OpenReview forum: "Improving Adversarial Defense with Self-supervised Test-time Fine-tuning"
_ICLR.cc/2022/Conference — ICLR 2022 Submitted_

### Official Review · Reviewer_itcW · 2021-11-01

**Correctness:** 3
**Technical Novelty And Significance:** 3
**Empirical Novelty And Significance:** 2
**Recommendation:** 3
**Confidence:** 4

**Main Review:**

Strengths:
========
Robust overfitting of adversarial training is a well known problem and thus require separate attention , in that context the paper is well motivated. The results are good in the sense that with Meta AT the accuracy improvement is consistent.

Weaknesses:
===========
1. I am not sure whether it is at all feasible to have the test samples even without labels before head to do self-supervised fine tuning. The assumption of fine tuning on test mini batch thus definitely limits the scope of applications of the proposed method.

2. Test time training (TTT) as a fine tuning for a full test set with the full model might be extremely in efficient and compute heavy that many inference platforms can't support.

3. Though understand what the variable is,  please be specific about the definition of $\epsilon$, the size of budget is technically not correct.

4. The difference that the authors mentioned from [1], looks like disadvantage to me, as the original paper did incremental fine tuning, whereas the current paper does fine tuning starting from same weights, for each image, The reasoning to take this different strategy is unclear. Also, if the batch size is >1, how does the author manage to start from same $\theta^0$ for each image of a batch? Please also mention what the $\mathcal{L}_{SS}$ loss is? For example, it is definitely not any cross entropy loss.

5. Also, please explicitly mention  the $\theta_{E}$ and $\theta_{f}$  definitions that are used for the multi task formulation.

6. How is the author sure about that the $\theta^0$ yielded by meta learner gives best starting point. I think the author meant near optimal, hence please rephrase your sentences.

7. Eq. 4 uses test samples $\hat{B}_j$ for SS-loss where as Eq. 7 uses training sample batch for  SS-loss, why?

8. Please do refer and compare with the paper [2] as they seem to solve similar issues of adversarial training.

9. Please provide a detailed analysis of the training and inference time and compute complexity over head due to the meta adversarial training and TTT.

10. How the authors use the same starting $\theta^0$ for batch size of $B$ > 1 is not clear to me. Please clarify.

11. The comparison with Regular AT and Regular AT + w/o FT in Table 4 is unfair to me, as in these cases the user has no chance to do any fine tuning with the test samples.


[1] Test-time training with self supervision for generalization under distribution shifts, ICML 2020.
[2] Robust Overfitting may be mitigated by properly learned smoothening, ICLR 2021.

**Summary Of The Paper:**

The paper introduces a meta adversarial training method to find a good starting point for test-time fine-tuning. It incorporates the test-time fine-tuning procedure into the training phase and strengthens the correlation between the self-supervised and classification tasks. Results on CIFAR-10, STL10 and Tiny-ImageNet show consistent robustness improvement.

**Summary Of The Review:**

The authors proposes a costly training and slow fine tune based testing strategy to get rid of robust overfitting issues associated with AT. In this context the authors should justify the increased cost of training/inference with gain in performance trade-off. The authors should also compare with at least the one relevant paper I mentioned.  I can comment on the overall incremental novelty once I get to know the purpose of doing few things differently. For example the difference between TTT an proposed solution is less, and I am not convinced the $\theta^0$ initiation is necessary always. Please justify.

---

> ### Author Response · Authors · 2021-11-22
> **Response to Reviewer itcW**
>
> Thanks for the constructive comments, and we clarify the concerns as follows:
>
> 1.**Setting of the test-time fine-tuning.** Our setting of online fine-tuning is practical. When the classifier needs to make predictions about one image, it has to see the input before making predictions. Our algorithm can perform online test-time fine-tuning just on this image without knowing its label. It does not have additional requirements on the number of data samples. Our method is different from TTT as we do not require the full test set to fine-tune the model.
>
> 2.**Difference between our method and TTT.** Our method and TTT focus on different scenarios about data distribution. TTT focuses on the problem of system distribution shifts. Therefore, $\theta$ needs to be changed for all the samples. Our method is focusing on the data distribution of large variance. $\theta^0$ corresponds to the center of the distribution. And for each instance, we need to adapt $\theta$ to this instance to achieve better performance. Therefore, we need to start from $\theta^0$ to perform test-time fine-tuning for each instance.
>
> 3.**Batch Size $b$>1.** We fine-tune the model’s parameter for the whole minibatch to one $\theta^T$ as shown in Algorithm 1. The prediction of all the samples in the minibatch is given by this $\theta^T$.
>
> 4.**Definition of** $\theta_E$, $\theta_f$ **and** $\epsilon$.  $\theta_E$, $\theta_f$ and $\epsilon$ are defined in the beginning of Section 3. $\theta_E$ means the parameter of the backbone $E$. $\theta_f$ if the parameter of the classification head and $\epsilon$ is the maximum norm of adversarial perturbation.
>
> 5.**Starting Point $\theta^0$.** We have rephrased the statement saying that $\theta^0$ is a better starting point than regular adversarial training.
>
> 6.**Eq.4 and Eq.7.** The test-time fine-tuning is to use test samples. So we use test samples to perform test-time fine-tuning in Eq. 4. Eq .7 refers to the Meta AT on the training data where only training samples are available. Therefore we use the training batch $B_j$.
>
> 7.**Comparison to [2].** Our work focuses on a different problem of adversarial training from [2]. Robust overfitting in [2] refers to the phenomenon that the model’s performance decays significantly on the test set in the later phase of adversarial training. [2] solves the problem with knowledge distillation and weight averaging. The overfitting mentioned in our work refers to the large training-test accuracy gap, which is different from [2].
>
> 8.**Training Time for Meta AT.** We compare the training time of Meta AT with regular AT without and with self-supervised losses. The following table shows the number of forward-backward passes of one step and the training time of one epoch for each method. It shows that the overhead of Meta AT is not very large.
>
> |                                 | Regular AT | Regular AT w/ Rotation and VFlip | Meta AT w/ Rotation and VFlip |
> | ------------------------------- | ---------- | -------------------------------- | ----------------------------- |
> | Number of  Passes               | 11         | 19                               | 24                            |
> | Training time for 1 epoch (min) | 20.75      | 32.92                            | 38.07                         |
>
> 9.**Time for Inference.**  The following table shows the average inference time for each instance using different methods. SOAP [1] is a previous method that purifies the adversarial example in the test time. While the inference time for our method is larger than the normal inference and SOAP when the batch size is 1, the inference time gets closer when using larger batch size.  And the batch size of 20 or more is a common scenari, the inference time of our method is similar to previous test-time fine-tuning methods.
>
>    | Batch Size       | 1      | 5      | 10     | 20     | 40     |
>    | ---------------- | ------ | ------ | ------ | ------ | ------ |
>    | Normal | 17.1ms | 14.5ms | 13.2ms | 12.8ms | 11.7ms |
>    | SOAP [1]         | 163ms  | 91.2ms | 75.3ms | 73.1ms | 72.5ms |
>    | Ours             | 545ms  | 168ms  | 118ms  | 83.9ms | 82.9ms |
>
> 10.**Comparison with Regular AT and Regular AT + w/o FT.** One contribution of our work is to use test-time fine-tuning to improve the robust accuray. Therefore, we need to use these methods as baselines to show the effectiveness of test-time fine-tuning. Regular AT and Regular AT + w/o FT are standard inference pipelines that directly take the input and output the prediction used in previous works of adversarial training. Besides, the test labels are not leaked during our test-time fine-tuning. Therefore, the comparison is fair.
>
> [1] Shi et.al, Online Adversarial Purification based on Self-supervised Learning, ICLR 2021
>
> [2] Chen et.al., Robust Overfitting may be mitigated by properly learned smoothening, ICLR 2021.

---

> > ### Comment · Reviewer_itcW · 2021-11-28
> > **Thanks for the rebuttal**
> >
> > Dear authors,
> >
> > Thanks for your efforts in writing a detailed rebuttal. However, unfortunately, I still think this is not a practically deployable solution, particularly, when other alternatives can resolve similar issues at much efficient training time. Also, the increase in training time with Meta is significant for it to be considered anywhere near efficient. Thus, I retain my score. I hope the author would work on improving the training strategy further to make it comparable to standard AT.

---

### Official Review · Reviewer_VL4k · 2021-11-02

**Correctness:** 4
**Technical Novelty And Significance:** 3
**Empirical Novelty And Significance:** 3
**Recommendation:** 6
**Confidence:** 3

**Main Review:**

In summary, there isn't much to critique about this paper. The exposition was clear, barring a number of typos/grammatical errors which should be resolved in the revision. While the white-box version of the adaptive attack was unconvincing due to the gradient estimate used, the authors supplemented these results with a black-box attack that does not rely on local gradient information. Overall, a well executed paper. My only remaining concern is w.r.t. related work, while a number of papers are listed below, a comparison to existing work leveraging test-time training for adversarial robustness would be warranted.

Test-time training applied to adversarial defense: https://arxiv.org/abs/2105.08714.
Explored addition of unlabeled data for adversarial training: https://arxiv.org/abs/2010.03593.
Applied self-supervised learning technique for robust learning: https://openreview.net/forum?id=bgQek2O63w

- Some typos, e.g. "As SGD is more efficient more large amount of data..."
- "...we use SGD to optimize $\theta$ when b is large (e.g. offline setting)." Please explicitly state what's done when b is small.
- "... strengthens the correlation between the self-supervised and classification tasks" what evidence was provided that meta-adversarial training strengthens the correlation between the self-supervised and classification tasks? The analysis showed that if these tasks are correlated, self-supervised fine-tuning will help, but I do not see where this particular statement in the abstract is supported.



**Summary Of The Paper:**

Propose to improve the generalization and robust accuracy of adversarially-trained networks via self-supervised test-time fine-tuning. Introduce a meta adversarial training method to find a good starting point for test-time fine-tuning, which  incorporates the test-time fine-tuning procedure into the training phase. Observe consistent improvement under different attack strategies for both white-box and black-box attacks.

**Summary Of The Review:**

Paper is well-written and lacks any obvious concerns regarding the content, other than the lack of comparison against existing methods for leveraging test-time training for adversarial defence.

---

> ### Author Response · Authors · 2021-11-22
> **Response to Reviewer VL4k**
>
> Thanks for the constructive review. We have fixed the typos in the revision.
>
>
> 1.**When $b$ is small.** When $b$ is small, we use the online fine-tuning algorithm shown in Algorithm 1. $b=1$ is the online test-time fine-tuning that we perform in the experiments.
>
> 2.**Correlation.** The objective function of Meta AT is
>
> $L_{cls}(B_j^{adv}; \theta-\alpha \nabla_\theta {L}_{SS}(B_j^{adv}; \theta))$
>
> The first order Taylor expansion with respect to $\theta$ gives
>
> $L_{cls}(B_j^{adv}; \theta) - \alpha \nabla_\theta L_{SS}(B_j^{adv}; \theta)^T \nabla_\theta L_{cls}(B_j^{adv}; \theta)$
>
> Therefore, minimizing the objective function is to improve the correlation between gradients of the self-supervised tasks and the classification task in the second term. In Table 5, we perform an ablation study comparing Meta AT and Regular AT. The test-time fine-tuning of Meta AT has a larger improvement than Regular AT. It shows that Meta AT is effective.

---

> > ### Comment · Reviewer_VL4k · 2021-11-22
> > **Response**
> >
> > Please add (1) and (2) to the revision. I stand by my previous statement that, in my view, this paper does not clearly pass the threshold for acceptance, as I don't have clarity on how this method compares to existing work for applying self-supervised learning at train time and/or test time for adversarial robustness.

---

### Official Review · Reviewer_tnn3 · 2021-11-03

**Correctness:** 3
**Technical Novelty And Significance:** 2
**Empirical Novelty And Significance:** 3
**Recommendation:** 3
**Confidence:** 4

**Main Review:**

Strengths
- **Optimization for Defense is a Worthy Topic**. Attacks optimize against defenses so it is entirely sensible that research should explore defenses that optimize back. The proposed method in particular addresses not just the test-time fine-tuning to do so, but a training-time preparation to do so by meta-learning.
- **Standard Experimental Design**. The choice of attacks comprise the standard AutoAttack benchmark. There are standard choices of dataset (CIFAR-10) and architectures (wide resnet, standard resnet) that are common for work on adversarial defense. The baselines are sensible, and separate the effects of adversarial training, multi-task training with self-supervision, and the meta-learning variant of adversarial training.
- **Method Variants**. The experiments cover different scopes, like online or offline optimization, and ablations of the loss components (self-supervision and regularization) and meta adversarial training. The different parts of the proposed method all help. Furthermore, offline optimization on the full test set improves over online optimization on a portion of it, which could be a positive if the fine-tuned model transfers to more attacked data without further optimization.
- **Popular Choice of Self-Supervision**: rotation prediction is a common and effective choice for use and benchmarking (the Gidaris et al. 2018 paper on it is cited 1000+ times).

Weaknesses
- **There are Prior Test-Time Defenses**. There is existing work on test-time optimization of the model parameters (runtime masking and cleansing ICML'20), of model latents (Schott et al. ICLR'19), and of the input (SOAP/Online Adversarial Purification based on Self-Supervision ICLR'21). The last method is especially related as it optimizes self-supervised tasks at test-time to improve robustness. Apart from this recent wave, there is prior work on randomization at test-time (Raff et al., 2019; Cohen et al., 2019; Pang et al., 2020, Dhillon et al., 2018) and projecting inputs onto generative model distributions (Song et al., 2018; Samangouei et al., 2018; Hill et al., 2021). Although it could be argued this work differs from most in optimizing the model parameters, and not the input or latent variables, the point is somewhat moot because the optimization is episodic so parameter updates do not persist and affect inference on other data. None of these prior methods are included as baselines, but at least SOAP should be, as it also optimizes self-supervised tasks during testing.
- **Computational Cost**. Optimization during inference is more intensive than a mere forward pass of a static method. Even more expensive still is joint optimization on batches of training data and test data, as done by this method. Other test-time defenses only require test data (SOAP, https://arxiv.org/abs/2103.14347, https://arxiv.org/abs/2105.08714), or small amounts of auxiliary data (https://arxiv.org/abs/2103.14222). This expense could be justified by the amount of robustness gained, but the ratio here is not reported.
- **Main Attacks are for Static Defenses**. AutoAttack is a strong suite of attacks, but it is designed with _static_ defenses in mind, and the corresponding RobustBench benchmark only includes defenses that are deterministic and do not optimize during inference. With this in mind, improving robustness on this benchmark is necessary but not sufficient to show that a test-time defense is better. While one adaptive attack is provided, more should be done to heed well-established warnings about obfuscated gradients (Athalye et al. ICML 2018). Possibilities include a transfer attack from the static adversarial defense, an expectation attack on one test input with several samples of training and testing batches, and evaluation with a decision-based attack like boundary.
- **Narrow Look at Self-Supervision**. Rotation prediction is a popular task, but it is just one. Vertical flip prediction is so closely related that it is not so informative as a second task. A generic task, like reconstruction/auto-encoding, or a state-of-the-art task, like contrastive learning, would be more informative for the utility of self-supervision for defense.

For Rebuttal
- Please discuss the contributions of this work relative to the cited test-time defenses, and SOAP in particular https://openreview.net/forum?id=_i3ASPp12WS. Ideally this discussion could compare the results for optimizing the input, like SOAP, vs. optimizing the model parameters, like this work.
- Please comment on the computational requirements of this defense. How much more expensive is it in time and FLOPs than a single forward pass?
- Please discuss the obfuscated gradients checklist from Athalye et al. 2018 and cover how this work cannot be explained away as gradient obfuscation.
- Please experiment with a decision based attack, such as the boundary attack.

Miscellaneous Feedback
- [related work] Alongside the already published references mentioned above, there are a number of contemporary works published or posted to arxiv this year, which this work could discuss and compare against to strengthen its scope and evaluation: https://arxiv.org/abs/2103.14222, https://arxiv.org/abs/2103.14347, https://arxiv.org/abs/2105.08714, https://arxiv.org/abs/2106.04938, https://openreview.net/forum?id=RFGkzxMFqby, https://openreview.net/forum?id=36rU1ecTFvR, https://openreview.net/forum?id=3Uk9_JRVwiF

**Summary Of The Paper:**

Fine-tuning the parameters of an adversarially-trained model at test-time with self-supervised tasks improves robustness to standard adversarial attacks. By incorporating meta-learning into the adversarial training stage, the test-time fine-tuning achieves marginally higher accuracy (though not reduced optimization time, or at least this is not reported). The test-time optimization approach requires the training data be kept available for joint updates on training batches and test data. The self-supervised tasks considered are rotation prediction, a common choice, and vertical flip recognition, which is a close relative of it. Results are reported on the standard adversarial defense benchmark of CIFAR-10, and additionally models are also evaluated on STL and Tiny ImageNet. Design choices are ablated and an adaptive attack is designed and experimented with, but there other checks against obfuscated gradients (for instance those suggested by Athalye et al. 2018) are not pursued, such as attacks in expectation, or decision-based attacks. No comment is made about the computational cost of test-time fine-tuning.

**Summary Of The Review:**

The contributions are summarized as (1) introducing the framework of test-time fine-tuning for defense, (2) meta adversarial training, and (3) the adaptive attack against the proposed defense. However, the first and most novel contribution (1) needs qualification, as there is substantial prior work on test-time optimization for defense, and none are included as baselines. This lack of discussion and experiment makes the work incomplete. (2) is indeed a contribution. On the other hand (3) is more a requirement for proposing a defense than an independent contribution. As outlined in the main review, more could be done here to evaluate that the defense is more robust and not superficially interfering with attacks. As there is a wave of such defenses at present, it is important to verify improvement on what has already been done, and to work toward common evaluations techniques across them. As such, I encourage the authors to further test the proposed defense, but must recommend rejection of this edition of the work.

**Final Review** The rebuttal responded to the weaknesses of computational cost and the need for more attack types to assess gradient obfuscation but only partially addressed the prior and concurrent work on test-time optimization for defense. On the plus side, the potential issue of gradient masking is resolved by evaluation of a decision attack and reporting that robustness fails as the adversarial budget increases. However, on the minus side the computational cost of the method is significant and the transfer attack reduces the improvement in robustness from fine-tuning. More importantly, the response does not acknowledge the prior method of runtime masking and cleansing, which likewise updates model parameters. Furthermore, because the parameter updates in this work are _episodic_, it is perhaps not as different from purification methods as claimed. More work is needed to situate the contributions with respect to prior defenses, such as analysing when the model parameters (for this method) or input perturbation (for purification methods) is shared/unshared across the batch.

---

> ### Author Response · Authors · 2021-11-22
> **Response to Reviewer tnn3**
>
> We thank the reviewer for the constructive comments.
>
> 1.**Comparison to Previous Works.** Our method is intrinsically different from different methods as we are fine-tuning the model to adapt to new examples instead of purifying the input. We apply SOAP-RP to the adversarially-trained model and find that its improvement  is marginal. Under AutoPGD, the accuracy is improved from 53.09% to 53.57%. This improvement is much smaller than our method, whose improvement is from 53.09% to 57.93%. Note that the 53.57% accuracy is already higher than 51.90% reported in their paper as we use the adversarially-trained model as the start point. SOAP only has little effect when combining with the commonly-used adversarial training.
>
> 2.**Time for Inference.** The following table shows the average inference time for each instance using different methods. While the inference time for our method is larger than SOAP when the batch size is 1, the inference time gets closer when using larger batch size. And the batch size of 20 or more is a common scenario of the inference, the inference time of our method is similar to SOAP.
>
>    | Batch Size       | 1      | 5      | 10     | 20     | 40     |
>    | ---------------- | ------ | ------ | ------ | ------ | ------ |
>    | Normal | 17.1ms | 14.5ms | 13.2ms | 12.8ms | 11.7ms |
>    | SOAP             | 163ms  | 91.2ms | 75.3ms | 73.1ms | 72.5ms |
>    | Ours             | 545ms  | 168ms  | 118ms  | 83.9ms | 82.9ms |
>
> 3.**Gradient Masking.** Following the methods of Athalye et al. 2018, we perform a transfer attack from the static adversarial defense and the expectation attack. For transfer attack, we use the robust networks with the same architecture as the substitute model, and the test-time fine-tuning also improves the robust accuracy.
>
> | Tasks           | Rotation | VFlip   | Rotation+VFlip |
> | --------------- | -------- | ------- | -------------- |
> | w/o Fine-tuning | 65.58 %  | 66.13 % | 66.83 %        |
> | w/ Fine-tuning  | 66.45 %  | 66.95 % | 67.97%         |
>
> For expectation attack, we modify the adaptive attack and average the gradient from 10 fine-tuned models, whose training batches are different. We evaluate the model using the ensemble of rotation and vertical flip as the self-supervised task on CIFAR10. Based on our limited computational resources, we only evaluate Adaptive-AutoPGD-EOT and Adaptive-SquareAttack-EOT. One is the strongest attack in our method and the other is a black-box attack that is less likely to be affected by gradient masking.
>
> |                           | w/o Fine-tuning | w/ Fine-tuning |
> | ------------------------- | --------------- | -------------- |
> | Adaptive-AutoPGD          | 53.99%          | 57.70%         |
> | Adaptive-AutoPGD-EOT      | 53.99%          | 57.65%         |
> | Adaptive-SquareAttack     | 65.75%          | 66.80%         |
> | Adaptive-SquareAttack-EOT | 65.75%          | 66.73%         |
>
> The experiment shows that the expectation attack has little influence on the improvement of our test-time fine-tuning.
>
> In addition, Table 7 in the appendix shows that our method also improves the clean accuracy when performing test-time fine-tuning. If our improvement is the result of gradient masking, the accuracy would not be improved on the clean data.
>
> 4.**Boundary Attack.** We use one of the SOTA decision-based attacks: RayS [1]. Based on our limited computational resources, we only test it on CIFAR10 with the ensemble of rotation and vertical flip. The accuracy in the following table shows that our method also improves the robust accuracy for the decision-based attack.
>
>    |                 | RayS   | Adaptive-RayS |
>    | --------------- | ------ | ------------- |
>    | w/o fine-tuning | 65.61% | /             |
>    | w/ fine-tuning  | 77.38% | 75.03%        |
>
> 5.**Checklist of Athalye et al. 2018.**
>    - For our method, black-box attacks like Square Attack and transfer attack achieve much higher accuracy than the strong white-box attack.
>    - For the unbounded attack, when increasing the adversarial budget to 1, the robust accuracy of our method will decrease to 0, which is the case for our method.
>
> [1] Chen et.al., RayS: A Ray Searching Method for Hard-label Adversarial Attack, KDD 2020

---

> > ### Comment · Reviewer_tnn3 · 2021-11-30
> > **Thank you for the clarification, timing measurements, and additional evaluation!**
> >
> > For rebuttal my review highlighted prior work, the need to know computational cost, and insufficient evaluation to rule out gradient masking. In brief, in considering the response the the coverage of prior work is still dissatisfactory, the computational cost is high, and the additional evaluation is convincing that gradient masking is not the main effect. I have maintained my score of 3, mostly due to the persistent concern about the situation of the contributions w.r.t. prior work.
> >
> > I considered raising the score to 5, but could not given the concern about prior work and the computational cost required. All the same, I thank the authors for improving the evaluation with the additional attack types considered, and encourage these to be included in the next version.
> >
> > > 1.Comparison to Previous Works. Our method is intrinsically different from different methods as we are fine-tuning the model to adapt to new examples instead of purifying the input.
> > > We apply SOAP-RP to the adversarially-trained model and find that its improvement is marginal.
> >
> > First, there is a prior method of the same class: Runtime Masking and Cleansing (see review) also fine-tunes the model parameters on testing data. Second, the importance of this "intrinsic" difference needs to be demonstrated by experiment. As mentioned in my review, because the model updates are episodic, the proposed method is more related to prior work that optimizes latents or the inputs. This is because these updates will not be shared across other inputs, nor alter the initial state of inference on new data, as persistent/non-episodic model updates would.
> >
> > Thank you for reporting a single run for SOAP. In fairness to the prior method, it may be necessary to try different optimization hyperparameters due to the change in base model, or to try the same combination of self-supervised tasks as in the proposed method (rotation and flipping).
> >
> > Results with SOAP and RMC should absolutely be included in this work. Otherwise it lacks baselines for methods of the same class as the proposed method: test-time defenses that update the model and/or input.
> >
> > > 2.Time for Inference.
> >
> > The time for inference should absolutely be reported in the paper. As the point of the method is to change inference, it is necessary to know how the computation of inference changes along with the robust accuracy. As it is, the cost of 50x or 8x slower inference relative to adversarial training depending on batch size is significant, so the paper should discuss the trade-off of computation and robust accuracy.
> >
> > > 5.Checklist of Athalye et al. 2018.
> >
> > These two points are indeed signs that obfuscation is not an issue, so please include them in the paper.

---

### Official Review · Reviewer_wsz9 · 2021-11-08

**Correctness:** 4
**Technical Novelty And Significance:** 3
**Empirical Novelty And Significance:** 3
**Recommendation:** 5
**Confidence:** 4

**Main Review:**

The test-time fine tuning method for better adversarial robustness performance is very interesting. Recently, the test-time training techniques have gained popularity mainly due to their improved performance and adaptability to different domains. Using this strategy for enhanced adversarial robustness is definitely a nice idea. While the authors presented the method with proper motivation followed by results that shows performance improvement, there are some points I would like to be clarified on as given below.

1)Basic methodology of the proposed work is mainly based on the papers [Hendrycks et al., 2019] (https://arxiv.org/pdf/1906.12340.pdf) and [Sun et al., 2020] (https://arxiv.org/pdf/1909.13231.pdf), the first one was not even mentioned anywhere in the introduction or the related work sections. While the improvements made over these mentioned papers are significant, it's very important to properly explain the previous efforts in this space and position the current work for a better understanding of the actual contribution.

2)The method analysis subsection (3.4) is not well explained in my view. It's not clear to me how a correlation mean, larger than robust accuracy of the adversarially trained network, implies the effectiveness of the proposed method.

3)While the results, presented for 3 different datasets, show good adversarial robustness performances against different attack methods, they were only compared with baselines that are other variations of the proposed method. It's very important to add results of other SOTA adversarial robustness methods that would improve the acceptance of the proposed method.

4)The only drawback of test-time training methods could be arguably the increased inference time. As adversarial training itself is a highly time-consuming method mainly for the inner maximization i.e. generating the adversarial images, it's important to show the time required for the proposed test-time fine tuning for adversarially robust methods.

**Summary Of The Paper:**

This paper presents a nice framework of test-time fine tuning through self-supervision for adversarially trained networks with the purpose of improving the robust accuracy on test data. A meta adversarial training strategy is also proposed that strengthens the correlation between the self-supervised and classification tasks to provide a good starting point for test-time fine tuning. The proposed method improved robustness performance against different attack methods including adaptive attacks where the attacker has knowledge of the fine tuning technique.

**Summary Of The Review:**

An interesting work that tried to improve adversarial robustness through test-time fine tuning using self-supervision techniques. Answering/commenting on the points, that I posted under the main review, can significantly improve the quality of this work.

---

> ### Author Response · Authors · 2021-11-22
> **Response to Reviewer wsz9**
>
> Thanks for the constructive review. We clarify the concerns as follows:
>
> 1.**Related works.** These papers are important works and we have cited these papers in our work. We have added more discussion about these papers in the related works.
>
> 2.**Section 3.4.** Section 3.4 analyzes the correlation of gradients between the self-supervised tasks and the classification task. Considering the first order Taylor expansion of the classification loss in Eqn (15).
> Larger correlation $\rho$ indicates smaller classification loss after fine-tuning. The positive correlation $\rho$ means that fine-tuning will decrease the classification loss  in average and it may make the wrong prediction to be correct. The phenomenon that the fraction of positive $\rho(\tilde{x}^{adv}_i)$ larger than the robust accuracy indicates the classification loss of more instances will decrease after fine-tuning and the accuracy will be improved.
>
> 3.**SOTA Baselines.** We select one SOTA adversarial training algorithm with additional data [1]. We apply our Meta AT to [1] with the rotation and vertical flip tasks. Using a WideResNet-28-10, it achieves robust accuracy **62.07%** under AutoPGD. With our test-time fine-tuning, the robust accuracy is improved to **64.34%**. The improvement of robust accuracy is **2.27%**. It shows that our method can be combined with SOTA adversarial training algorithm to further improve its robust accuracy.
>
> 4.**Time for Inference.**  The following table shows the average inference time for each instance using different methods. SOAP [2] is a previous method that purifies the adversarial example in the test time. While the inference time for our method is larger than the normal inference and SOAP when the batch size is 1, the inference time gets closer when using larger batch size.  And as the batch size of 20 or more is a common scenario, the inference time of our method is similar to previous test-time fine-tuning methods.
>
> | Batch Size | 1      | 5      | 10     | 20     | 40     |
> | ---------- | ------ | ------ | ------ | ------ | ------ |
> | Normal     | 17.1ms | 14.5ms | 13.2ms | 12.8ms | 11.7ms |
> | SOAP [2]   | 163ms  | 91.2ms | 75.3ms | 73.1ms | 72.5ms |
> | Ours       | 545ms  | 168ms  | 118ms  | 83.9ms | 82.9ms |
>
> [1] Gowal et al., Uncovering the Limits of Adversarial Training against Norm-Bounded Adversarial Examples, arXiv preprint arXiv:2010.03593, 2020
>
> [2] Shi et.al, Online Adversarial Purification based on Self-supervised Learning, ICLR 2021

---

### Decision · Program_Chairs · 2022-01-20

**Decision:**

Reject

**Comment:**

This paper proposes a method to improve the robust accuracy of classifiers using test-time training.  The reviewers all agree that the method is interesting, and many reviewers had a positive view of the method.  However, two main criticisms remain: (i) the method increases the runtime of inference, and (ii) comparisons to other related methods were lacking.  The authors responded to (i) by reporting runtimes for their method in the rebuttal.  Some reviewers were concerned that the runtime increase of the method is not acceptable, however I am not very concerned with this issue since I think the paper contains an interesting methodology even if it’s not ready for deployment at the industrial scale.  However, issue (ii) does not seem to have been adequately addressed.  The comparison to SOAP is a welcome addition the reviewers acknowledge, but a number of other methods, for example masking and cleansing, are closely related (but different) and so comparisons should be provided.